# Addressable and adaptable intercellular communication via DNA messaging

John P. Marken [1] ✉ & Richard M. Murray [1]

Engineered consortia are a major research focus for synthetic biologists because they can implement sophisticated behaviors inaccessible to single-strain systems. However, this functional capacity is constrained by their constituent strains' ability to engage in complex communication. DNA messaging, by enabling information-rich channel-decoupled communication, is a promising candidate architecture for implementing complex communication. But its major advantage, its messages' dynamic mutability, is still unexplored. We develop a framework for addressable and adaptable DNA messaging that leverages all three of these advantages and implement it using plasmid conjugation in E. coli. Our system can bias the transfer of messages to targeted receiver strains by 100- to 1000-fold, and their recipient lists can be dynamically updated in situ to control the flow of information through the population. This work lays the foundation for future developments that further utilize the unique advantages of DNA messaging to engineer previously-inaccessible levels of complexity into biological systems.

A major current focus of synthetic biology research is to expand beyond the field's original paradigm of engineering a single cell strain for a particular application and to instead engineer consortia, which are populations consisting of multiple distinct cell types[1,2]. By enabling the division of labor among its constituent strains, a consortia-based approach allows each strain to specialize itself to its assigned task while minimizing the metabolic burden on itself[3]. Engineered consortia are, therefore, able to achieve higher levels of functional complexity[4–6] and evolutionary stability[7,8] than analogous single-strain systems.

In order for an engineered consortium to function properly, however, it is necessary that each of its constituent strains can stably coexist and act in concert with each other. This coordinated activity is maintained by intercellular communication systems that allow the strains to dynamically instruct each other to perform programmed functions, like modulating their growth rate or activating a target gene. The achievable complexity of a consortium's behavior is therefore constrained by the capacity of its communication channels to transmit complex messages[6]. Realizing this, the synthetic biology community has placed much effort towards expanding the toolbox of intercellular communication channels and enabling increasingly information-dense communication between cells[9–15].

These efforts have almost exclusively focused on a molecular architecture that we will term small molecule actuated communication (SMA communication), wherein a sender cell synthesizes a small molecule that diffuses through the extracellular environment to enter a receiver cell that contains the requisite machinery to initiate a preprogrammed response to the signal. SMA communication channels were originally implemented using molecular parts co-opted from quorum sensing systems[9], but in recent years the toolbox has expanded to include metabolites[10], hormones[13], and antibiotics[16,17] as signal vectors.

An alternative molecular architecture, DNA messaging, was proposed in a pioneering report by Ortiz and Endy[14]. Here, horizontal gene transfer mechanisms are co-opted into a communication channel that transmits DNA-encoded messages between cells (Supplementary Fig. 1). Because the actual content of the message is an arbitrary genetic sequence within the mobile vector itself, Ortiz and Endy coined the term "message-channel decoupling" to describe the fact that a single DNA-based communication channel can send different messages that contain different types of instructions to the recipient cells[14]. In contrast, SMA communication channels exhibit message-channel coupling because the nature of the encodable message is tied to the molecular identity of the signaling molecule. A homoserine

[1]Division of Biology and Biological Engineering, California Institute of Technology, Pasadena, CA, USA. ✉e-mail: jmarken@caltech.edu

lactone, for example, can only be used to encode the instruction to activate its cognate transcription factor, and an antibiotic can only be used to encode the instruction to kill its susceptible cell strains (Fig. 1a).

A second important advantage of DNA communication is that a single DNA message can encode a large amount of information content, as many horizontal gene transfer systems can easily transfer several kilobases of arbitrary sequence[18–20]. In contrast, SMA channels can only modulate their activity via the concentration of their signal vector, a single small molecule. This heavily constrains the information density of the message to the point where in applications like digital computation, where concentrations are interpreted binarily as either OFF or ON, a single SMA channel can only transmit a single bit of information[21].

Together, these advantages suggest that DNA messaging is an ideal communication architecture for engineering complex consortia with sophisticated information processing requirements. But although the ten years since the Ortiz–Endy report have seen an increased use of horizontal gene transfer systems by synthetic biologists to engineer environmental microbiomes in the gut or soil[22–24], further studies of such systems' ability to act specifically as a communication framework for engineered consortia have only been performed computationally[25–27]. Thus, to our knowledge, the original Ortiz–Endy report remains the only experimental usage of DNA-based communication to date.

Why is the case? One reason is that, though it was pioneering in its foresight, the Ortiz–Endy implementation did not demonstrate a third property of DNA communication that is critical in enabling the implementation of qualitatively new functionalities—the dynamic mutability of DNA messages. Unlike SMA channels, where the message is encoded into the structure of an immutable signal molecule, cells have the ability to express DNA editors that can make targeted changes to the content of the message in situ (Fig. 1b). This ability has only expanded with the recent explosion in research on programmable DNA editors like CRISPR–Cas systems, integrases, and base editors[28,29]. Although theoretical reports have rightly identified mutability as a key advantage of DNA messaging[25], to date, this property has not been experimentally demonstrated.

We, therefore, set out to develop a general and scalable architecture for DNA messaging that allows users to fully take advantage of all three of its unique properties: message-channel decoupling, high information density, and dynamic message mutability. In order to ensure our framework's compatibility with arbitrary messages transferred along arbitrary horizontal gene transfer systems, we used channel-orthogonal molecular tools to implement a functionality that is required in all communication systems—the ability to address the message to a targeted set of recipients.

Our addressing framework uses CRISPR–Cas systems to internally validate each message transfer event within the consortium, enabling the targeted delivery of a given message to any subset of the strains in a population. We additionally designed a framework for using integrases to modularly update messages' recipient lists in situ, enabling the control of information flow through a population. This work establishes a universally applicable framework for effective DNA-based communication that sets the stage for future efforts that expand its ability to implement previously-inaccessible functionalities into engineered consortia.

## Results

### Incorporating massage addressability into a plasmid conjugation-based communication system

We first describe the implementation of an addressability system for our DNA messaging framework. Any such implementation requires a means for the molecular recognition of specific genetic sequences, and we chose to use the CRISPR–Cas adaptive immunity system due to its ability to programmatically target and cleave desired nucleotide sequences on genetic vectors entering the cell[30,31]. Although multiple different Cas systems have been demonstrated to cleave and degrade DNA vectors within cells[32,33], we specifically chose to use the *S. pyogenes* Cas9 endonuclease system because it contains the required binding, unwinding, and cleaving activities within a single protein, facilitating its use in many different host organisms[34]. Additionally, well-developed procedures exist for generating large libraries of orthogonal single-guide RNAs (gRNAs) for the Cas9 system[15,35,36], and the small footprint of the gRNA binding site (23 bp) means that many such sites can be incorporated onto a DNA message without significantly burdening any potential sequence length constraints from the transfer system. Together, these properties make the Cas9–gRNA system an ideal candidate for implementing a scalable, modular, and host-orthogonal addressing system for DNA messaging.

The design of our addressability framework is as follows. Each receiver cell in the consortium expresses both Cas9 and a unique gRNA that serves as a molecular signature encoding its strain identity. The sender cells themselves require no additional molecular machinery, but the DNA message must contain an array of gRNA binding sites that

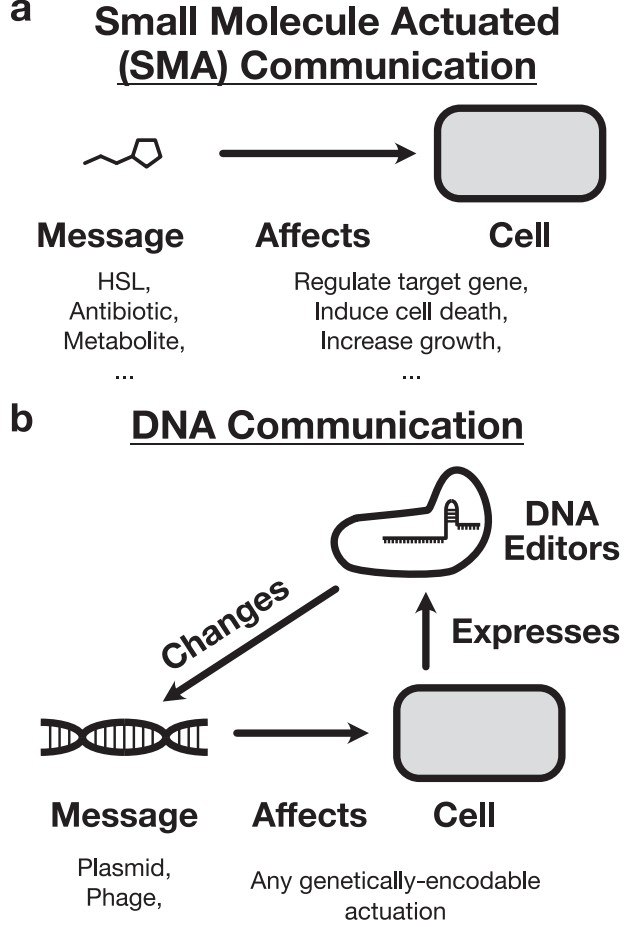

**a** Small Molecule Actuated (SMA) Communication

Message → Affects → Cell

HSL,
Antibiotic,
Metabolite,
…

Regulate target gene,
Induce cell death,
Increase growth,
…

**b** DNA Communication

DNA Editors

Changes · Expresses

Message → Affects → Cell

Plasmid,
Phage,
…

Any genetically-encodable actuation

**Fig. 1 | Architectures for engineered intercellular communication. a** Small molecule actuated (SMA) communication systems exhibit message-channel coupling, meaning that the behavior they induce in the receiver cell is hard-coded into the molecular identity of the signaling molecule itself. This molecular identity cannot be changed without disrupting the functioning of the channel itself. **b** DNA communication systems exhibit message-channel decoupling, meaning that a given channel can transmit multiple types of messages to induce any genetically-encodable response in the receiver cell. Furthermore, the cells themselves can express molecular DNA editors to change the content of the messages in situ, closing the loop to enable autonomous system reconfiguration.

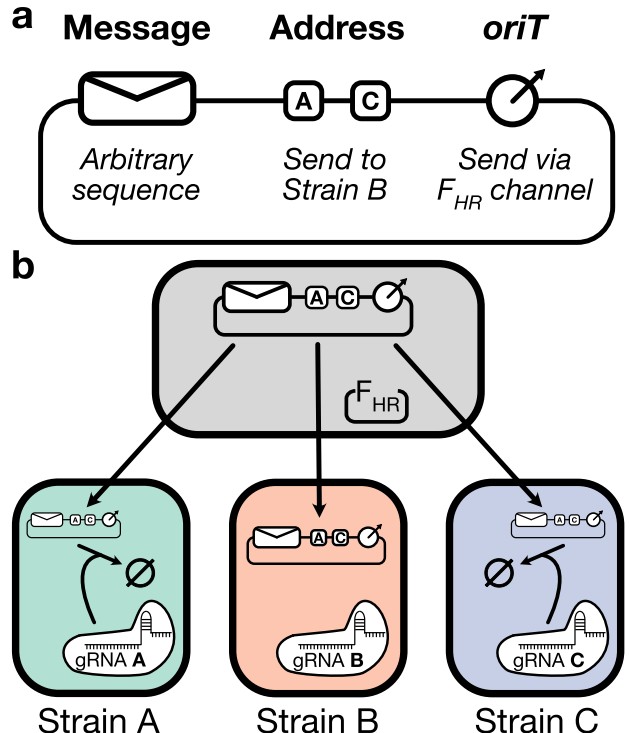

**Fig. 2 | Addressable DNA messaging. a** Schematic of an addressable DNA message. The content of the message is an arbitrary genetic sequence, and the address region uses gRNA binding sites to act as a blocklist that determines the message's recipient list by excluding transfer to all encoded strains. The origin of transfer (*oriT*) allows the message to interact with the cognate horizontal gene transfer machinery in the sender cell. **b** Schematic of transfer blocking. The DNA message is initially transferred promiscuously to all receiver strains in the population. As the message enters a receiver cell, the binding sites on the address region become exposed to cleavage by the Cas9–gRNA complex expressed within the cell. This cleavage only occurs if a binding site on the address matches the gRNA expressed in the receiver cell, thus ensuring that the message only persists within its appropriate recipients by eliminating the messages sent to invalid recipients.

correspond to the receiver strains that should not receive the message. We will refer to this array as the address region because it acts as a blocklist, encoding the recipient list of the message as the set of strains whose gRNAs are not encoded in the address (Fig. 2a). When the message is transferred to a receiver cell, the Cas9–gRNA complex checks the validity of the transfer—if the transfer is invalid, then the complex will bind to the cognate site on the address region and cleave the message, leading to its degradation. If the transfer is valid, then the Cas9–gRNA complex is unable to interact with the message, and so the message freely propagates within the receiver cell. This process is schematized in Fig. 2b.

An important property of this addressing framework is that the transfer validation system interacts with the message itself rather than the transfer machinery that carries the message. This means that a single DNA channel can send messages that are addressed to different recipients. When addressability is implemented via channel-intrinsic properties, such as in the Ortiz–Endy system's reliance on the M13 bacteriophage's narrow infection host range[14], every message that is transmitted by a channel must go to the same recipient list regardless of its content.

In demonstrating the incorporation of our message-addressing framework into a DNA-based communication system, we chose to deviate from Ortiz and Endy's original choice of the filamentous bacteriophage M13 and instead used a plasmid conjugation-based communication system. This is because the properties of plasmid conjugation systems are better aligned with the advantages of DNA-

based communication as a whole—plasmids can encode larger messages, with conjugative plasmids regularly reaching lengths of hundreds of kilobases[20,37], and can transfer to taxonomically-diverse recipients[38,39], facilitating their use in multispecies consortia. We specifically chose to use the $F_{HR}$ system developed by Dimitriu et al.[18], which is based on the *Escherichia coli* fertility factor F, the canonical representative of conjugative plasmids[40].

## Cas9-mediated blocking of plasmid receipt is inducible and orthogonal

In order to demonstrate that Cas9-mediated cleavage can indeed block the receipt of a mobilized plasmid, we performed pairwise sender-receiver experiments in *E. coli* consortia using the $F_{HR}$-based communication system. Receiver cells containing a genomically integrated spectinomycin resistance cassette were transformed with a plasmid encoding OHC14-HSL-inducible expression of Cas9 and one of two gRNAs ("A" or "B"), and sender cells containing a genomically integrated apramycin resistance cassette were transformed with the $F_{HR}$ helper plasmid and a pSC101 message plasmid that constitutively expresses a yellow fluorescent protein and chloramphenicol resistance gene. Two variants of this message plasmid were constructed, differing in whether their address region contained a single A binding site or a single B binding site (Fig. 3a).

With this setup, selective plating could be used to individually isolate the senders, receivers, and transconjugants from a mixed population and calculate their densities. We performed mating experiments on all four combinations of sender-receiver pairs in the presence and absence of OHC14-HSL induction and measured the densities of each strain after 6 h of growth in a shaken LB coculture (Fig. 3b). The message plasmid was transferred efficiently to the receivers in this timeframe, with an average of 64% of receivers being converted to transconjugants across all transfers to on-target recipients (Supplementary Fig. 2).

We then quantified the effectiveness of Cas9-mediated plasmid blocking by calculating the plasmid transfer rate in each experiment, defined as the transconjugant density divided by the product of the total sender and total receiver densities. We observed that when the Cas9 system was induced, the A-containing message plasmid had a 185-fold higher transfer rate to its valid recipient (the B receiver) than to its invalid recipient (the A receiver) ($p = 0.03$, paired $t$ test), and that for the B-containing message plasmid, the difference was 520-fold ($p = 0.01$, paired $t$ test) (Fig. 3c). When the Cas9 system was not induced in the receiver cells, this biased transfer was not observed ($p = 0.28$, 0.94, paired $t$ test, for A and B message plasmids, respectively) (Fig. 3c).

Having demonstrated that our addressability system performed successfully in a two-strain population, we next asked whether our system could scale to multi-strain populations where a given address region may need to encode several gRNA binding sites. We constructed three different receiver strains that, in addition to the spectinomycin resistance gene, each express a distinct fluorescent protein (mScarlet-I[41], sfYFP[42], or TagBFP[43]) from a genomically integrated cassette. In this way, all three receivers could be mixed together with the sender strain in a four-strain coculture, and the colors could be used to determine the density of each distinct receiver strain after selective plating. In order to further assess the generality of our Cas9-mediated blocking system, we used a set of orthogonal gRNAs developed by Didovyk et al.[35] instead of reusing the A and B gRNAs from the previous experiment. We transformed each of the colored receiver strains with a plasmid encoding Cas9 and one of three of the Didovyk gRNAs (D1, D2, or D3) and constructed sender strains containing one of eight message plasmids addressed to every possible combination of the three receiver strains (Fig. 4a).

We found that even in the more complex setting of a four-strain population, our system was able to preferentially deliver the message

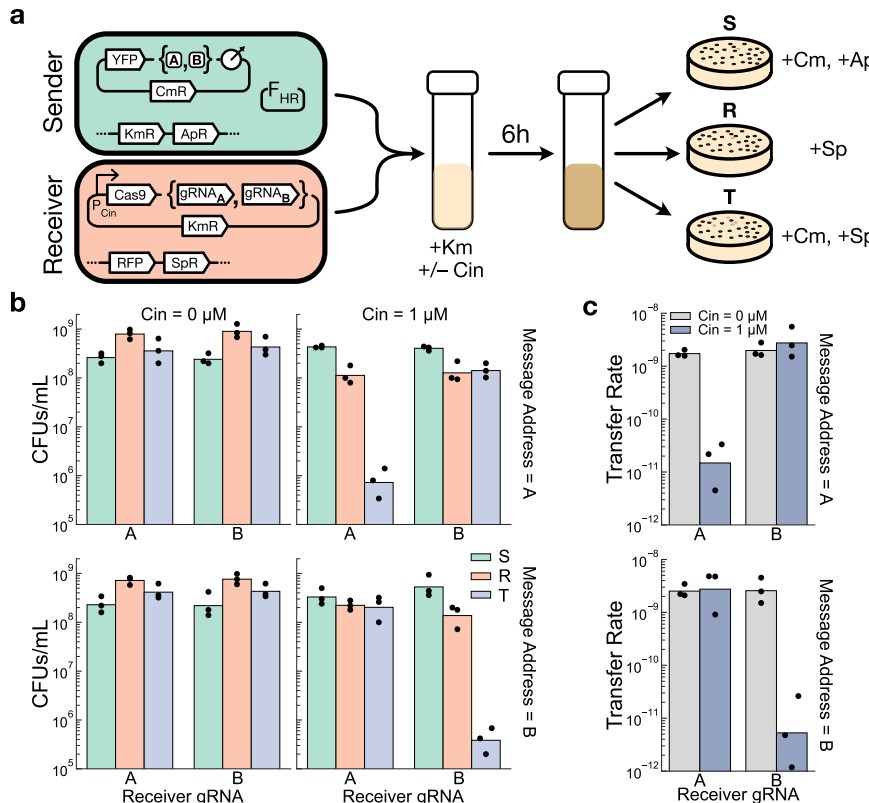

**Fig. 3 | Cas9-mediated cleavage of incoming plasmids can bias their transfer to targeted recipients. a** Schematic of the experimental setup. Senders (*S*) and Receivers (*R*) carrying one of two plasmid variants are grown together in a coculture, and selective plating is used to isolate them, as well as the transconjugants (T), from the mixed culture. Note that transconjugants will appear on the receiver-selecting plates, so *R* is the total density of receivers in the population (Methods).

**b** Endpoint strain densities, measured in colony forming units (CFUs) per mL of culture. (**c**) Transfer rates, calculated as $T/(S * R)$, of the message plasmid in each of the conditions in (**b**). Dots show the values from three biological replicates measured on different days, and bars depict the geometric mean of these values. Km kanamycin, Cm chloramphenicol, Ap apramycin, Sp spectinomycin, Cin = OHC14-HSL. Source data are provided as a Source Data file.

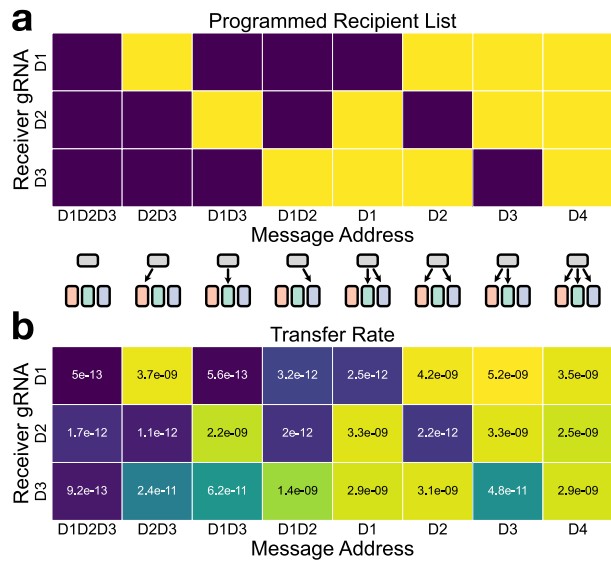

**Fig. 4 | Programmable delivery of message plasmids to arbitrary subsets of a multi-strain population. a** Schematic representation of the intended recipient list for each of the eight message plasmids. Dark squares indicate an invalid transfer, and light squares indicate a valid transfer. **b** The observed geometric mean of the transfer rates to each receiver type, calculated from three biological replicates measured on different days. The color map is scaled logarithmically over four orders of magnitude. Individual transfer rate values are shown in Supplementary Fig. 3. Source data are provided as a Source Data file.

to its appropriate recipients. Across all transfers to on-target recipients, the average fraction of receiver cells converted to transconjugants was 60%, and the fold change in transfer rate between valid and invalid recipients was often over 1000-fold (Fig. 4b; Supplementary Fig. 3). Although the three gRNAs used in the receivers were previously reported to be of comparable effectiveness in a dCas9-mediated transcriptional repression assay[35], the D1 and D2 gRNAs were able to block invalid transfers much more strongly than the D3 gRNA—the geometric mean of the fold change in transfer rates between valid and invalid recipients across all conditions where the invalid recipients expressed the D3 gRNA was 79-fold, compared to 1256-fold and 1577-fold for the D1 and D2 gRNAs, respectively (Supplementary Fig. 3).

### Cells can use integrases to edit DNA messages in situ and update their recipient list

Having demonstrated that our Cas9-mediated blocking system can successfully implement high-fidelity addressable communication between cells, we next proceeded to incorporate adaptability into the message transmission framework by enabling the programmable in situ editing of a message's recipient list. This can be accomplished by applying molecular DNA editors to modify the gRNA binding sites on the address region. Specifically, a system for programmable address editing should have the ability to both add a new binding site to the array and remove (or invalidate) an existing binding site from the array.

Serine integrases are a class of proteins that are well-suited for this task because of their ability to bind to specific attachment sequences and add, remove, or swap the regions between these sites depending on their configuration and orientation along the DNA[44,45]. Their

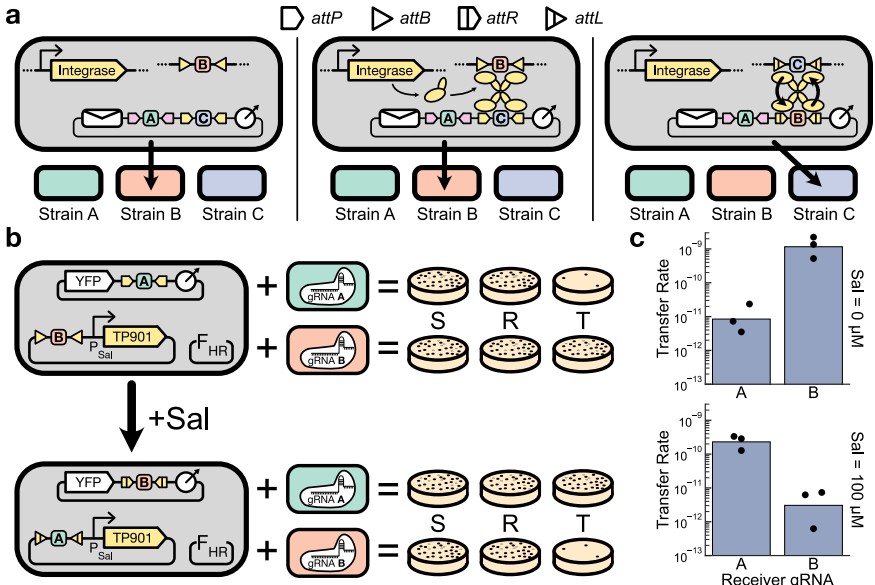

**Fig. 5 | Integrase-mediated address editing. a** Schematic of the process. In the left panel, expression of the integrase has not been induced, and no editing has occurred. The message is addressed to Strain B. Orthogonal integrase attachment sites flank each binding site on the address region. In the middle panel, the integrase associated with the C site on the address region has been induced. The corresponding attachment sites for this integrase are also encoded separately on a sequence distinct from the message plasmid. In the right panel, the cassette exchange process has been completed, and the C site on the address region has been swapped with a B site, updating the message's recipient list to Strain C. The process is unidirectional as it converts the attB and attP sites into attL and attR sites that can no longer undergo exchange, making this change permanent unless the cognate reverse directionality factors are expressed to reverse the process and restore the original sequence configuration[45]. **b** Experimental schematic. A single sender strain containing an address-editable message plasmid is coupled with one of two receiver strains in pairwise transfer experiments. Prior to editing, the message is addressed only to the B receiver, but after editing, the message is addressed only to the A receiver. **c** Measured transfer rates from the experiment described in (**b**). Dots show the values from three biological replicates measured on different days, and bars depict the geometric mean of these values. Sal salicylate. Source data are provided as a Source Data file.

efficiency and programmability have made their use ubiquitous among both molecular and synthetic biologists, and large sets of diverse and orthogonal integrases have been characterized[46,47].

We implemented address editing by flanking each binding site on the address region with orthogonal integrase attachment sites in such a way that the expression of the cognate integrase will swap the binding site with a different binding site contained on a separate non-mobile plasmid via a process called recombinase-mediated cassette exchange[48] (Fig. 5a). This procedure leaves the rest of the message, including the other binding sites on the address region, unaffected.

An important property of this address editing system is that it can be executed unilaterally by the sender cell, such that a message's recipient list can be updated without any coordination with the receivers themselves. This feature is once again only possible because our framework encodes a message's recipient list on the message itself rather than relying on channel-specific interactions between the message vector and the recipient cell.

To assess the efficacy of our address editing system, we constructed a single sender strain that contains a nonmobilizable plasmid encoding a salicylate-inducible TP901 integrase expression cassette and the B gRNA binding site flanked by TP901 attB sites alongside a message plasmid containing the A binding site flanked by TP901 attP sites in its address region. We then performed pairwise sender–receiver mating experiments on each of the two A- or B-expressing receiver strains from the original pairwise addressing experiments (Fig. 3), with the blocking system induced, in the presence or absence of salicylate induction (Fig. 5b). Following expectations, transfer of the message to the A receiver was blocked 138-fold in the absence of integrase activity ($p = 0.03$, paired $t$ test) while the blocking profile was reversed when the integrase was induced, with the transfer to the B receiver now being blocked by 75 fold ($p = 0.01$, paired $t$ test) (Fig. 5c).

Although the integrase system was successfully able to bias the transfer of the message plasmid to its intended recipient, we noticed that the overall efficiency of the transfer was lower, with an average receiver conversion of 30% in the pre-edit unblocked transfer dropping to 4.4% in the post-edit unblocked transfer (Supplementary Fig. 4a). In order to determine whether this decrease was due to a change in strain growth dynamics or a change in the intrinsic transfer rate of the plasmid, we compared the transfer rates from the above experiment with the original pairwise transfer blocking experiments in Fig. 3. We found that these data were nearly directly log-linearly related with the values obtained in the address editing experiment in all conditions except for the post-edit valid transfer (Supplementary Fig. 4b). This suggests that there was a global 2-fold decrease in the transfer rate of the Fig. 5 experiments compared to the Fig. 3 experiments, that was further exacerbated by an additional 5-fold drop in the post-edit on-target transfer. It is, therefore, prudent to note that the integrase-mediated address editing process can reduce the transfer rate to the new recipients, and mitigating this effect will be an important part of future optimizations of this DNA messaging system.

## Address editing enables control of information flow through a population

Having demonstrated that integrase-mediated cassette exchange can successfully edit the address region of a message plasmid to bias its transfer towards new recipients, we next used address editing to implement a proof-of-concept demonstration of controlled information flow through a population. Such control involves not only the selection of specific recipients for a message, as has already been demonstrated, but also the enforcing of a defined order for visiting these recipients. Enforced ordering is an essential part of coordinating multi-step processes, and its implementability is, therefore, a desirable property for intercellular communication systems.

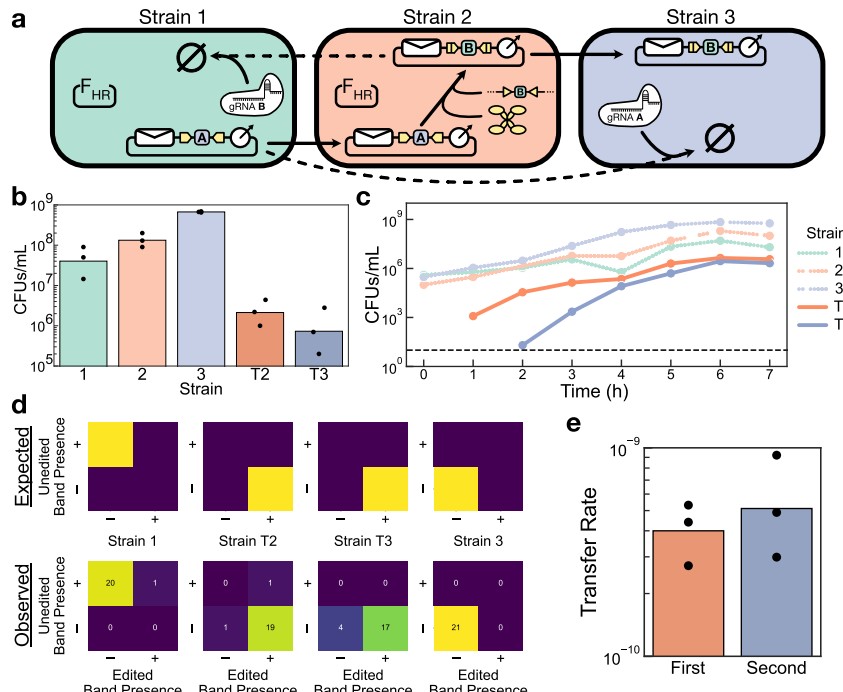

**Fig. 6 | A three-strain linear message relay. a** Schematic of the relay system. The message plasmid starts in Strain 1 and can only be transferred to Strain 2. When the message enters Strain 2, its address is edited so that it can no longer return to Strain 1 but is now allowed to continue on to Strain 3. This architecture is scalable to $n$ strains (Supplementary Fig. 5). **b** Endpoint densities of each strain after 6 h of coculture. **c** Timecourse plating of each strain within the coculture for a single biological replicate. The dashed black line marks the limit of detection. **d** PCR assays of endpoint colonies from selected strains. (Top row) The expected plasmid types in each strain are based on the relay design. (Bottom row) Results of the PCR assay from 21 colonies of each selected strain. The number of colonies that were assigned to each result condition is indicated in the heatmap. **e** Transfer rate calculated for each step in the relay, based on the data from (**b**). For the first transfer, Strain 1 is the sender, and Strain 2 is the receiver. For the second transfer, Strain T2 is the sender, and Strain 3 is the receiver. In all bar graphs, dots show the values from three biological replicates measured on different days, and bars depict the geometric mean of these values. Source data are provided as a Source Data file.

Specifically, we designed a linear message relay that forces the message to propagate in a linear sequence along a defined order of strains in a consortium without skipping ahead or backtracking (Fig. 6a). This sequential order is enforced by ensuring the message plasmid is only addressed to the next strain in the sequence at any given time, which can be implemented by having each successive strain edit the message's address accordingly. An important but subtle property of this system is that the entire signal relay is implemented using a single communication channel that modifies its message at each step. Implementing a similar signal relay with SMA channels would require $n-1$ orthogonal channels for an $n$-strain population, while the DNA-based implementation requires only a single channel regardless of the complexity of the consortium composition (Supplementary Fig. 5).

We designed the strains for a three-strain linear relay, as described in Fig. 6a—Strain 1 contains $F_{HR}$ and the message plasmid (initially addressed only to Strain 2) as well as the B gRNA, while Strain 2 contains $F_{HR}$ and the machinery to edit the message plasmid, when received, to address only Strain 3. Strain 3 itself simply expresses the A gRNA. A unique antibiotic resistance gene was genomically integrated into each strain (gentamicin, apramycin, and spectinomycin, respectively) to enable its selective isolation. We then mixed the three strains together and cocultured them for 6 h before selectively plating out each strain (three parent strains and two possible transconjugant strains) to measure their densities in the endpoint population state.

Because Strain 3's transconjugants (T3) cannot appear until Strain 2's transconjugants (T2) appear, we anticipated that the final endpoint density of T3 should be lower than that of T2. We observed that the average endpoint density of T3 was indeed 3-fold lower than that of T2, although not significantly so ($p = 0.09$, paired $t$ test). This effect emerged despite the fact that the average density of the T3 parent strain (Strain 3) was 5-fold higher than that of the T2 parent strain (Strain 2) ($p = 0.02$, paired $t$ test) (Fig. 6b). In general, the slow growth of the sender strains contributed to very low receiver conversion rates within the 6-h window (1.6% conversion of Strain 2 and 0.1% conversion of Strain 3), but these could be mitigated by designing the strains to have similar growth dynamics.

To further validate our system's ability to enforce the sequentially ordered transmission of its message, we performed a timecourse assay for one of our replicates where we plated out the coculture every hour after the initial mixing to obtain the growth curves of each strain over the course of the experiment (Fig. 6c). These results are again consistent with the desired system behavior, as Strain 3's transconjugants (T3) do not appear until after Strain 2's transconjugants (T2) have become detectable.

Finally, in order to confirm that each strain only contains the desired form of the message plasmid, we performed PCRs on colonies from the endpoint cultures that selectively amplified either the edited or unedited form of the message plasmid. The results aligned almost directly with the expectation for each tested strain, with the exception of four T3 colonies showing sub-threshold amplification of both plasmid types (Fig. 6d).

Having demonstrated that our system successfully enforced the sequential linear transfer of the message plasmid through the three-strain consortium, we next asked whether the act of editing the message plasmid imposed a penalty on its transfer rate as it did in Fig. 5. Encouragingly, we found that the rate of the second (post-edit) transfer in the relay was not lower than the rate of the first (pre-edit) transfer ($p = 0.61$, paired $t$ test) (Fig. 6e). One possible reason for this discrepancy is that in the relay system, the cell performing the editing

of the message plasmid only sees one message plasmid at a time as it enters the cell via conjugation, while in Fig. 5 the editing was activated in a cell where the message plasmid had already reached its steady-state copy number. Nevertheless, the fact that editing the message plasmid did not impose a detectable decrease in its transfer rate supports the scalability of this linear relay architecture to larger numbers of strains.

Taken together, these results confirm that the address editing system can indeed be used to reliably control the flow of messages through a population.

## Discussion

In this work, we have designed a modular, scalable, and adaptable message-addressing framework for DNA-based communication channels and implemented it in an F-mediated plasmid conjugation system in *E. coli* consortia. Because our addressing system is built with molecular components that are orthogonal to the native horizontal gene transfer machinery, any existing DNA-based communication channel can be modified to incorporate our addressing system by expressing Cas9 and a strain-identifying gRNA in the receiver cells and encoding an address region onto the message vector.

Because our goal was to provide a proof-of-concept demonstration of an adaptable DNA communication system, there are many fruitful directions for further optimization of this framework. For example, because we expressed our Cas9 and gRNA from a plasmid in our receiver cells, it is likely that mutation and plasmid loss created a subpopulation of receivers without a functional transfer blocking system[49,50]. By promoting its evolutionary stability, for example by integrating it onto multiple sites on the genome, it is possible that we could improve the system's ability to block off-target transfers even further.

Another promising direction is to improve and augment the transfer properties of the original horizontal gene transfer system itself. $F_{HR}$, like the M13 helper system, constitutively expresses its transfer machinery, but the master transcriptional regulators for these operons have been identified, and so could be engineered to increase their expression or place them under inducible control[51]. Interfacing more with the system's channel-intrinsic properties, for example, by modulating the expression of entry exclusion proteins to globally block plasmid receipt[52], could also add an additional layer of programmable functionality to the system.

Converting additional horizontally mobile genetic vectors into new DNA-based communication channels will also be an important component of the continued development of DNA messaging. For example, the F plasmid is known to stop conjugation as the population approaches stationary phase, which limits its overall transfer rate in liquid culture experiments[53]. We observed that this property could also hold for plasmids mobilized by $F_{HR}$ (Supplementary Fig. 6) and that this leads to a low overall transfer rate—only around 50% of the receivers in our pairwise transfer experiments were converted to transconjugants after 6 h of coculture when transfer blocking was not induced (Supplementary Fig. 2). In contrast, Ortiz and Endy were able to achieve over 90% receiver conversion after 5 h of coculture using their M13 bacteriophage-based system[14], despite the fact that the M13 transfer rate has been estimated to be lower than the F transfer rate in coculture conditions[54]. As different applications will be best served by systems with different transfer properties, developing a diverse and well-characterized toolbox of DNA communication channels will be important in facilitating their wider use.

One potential class of applications where the use of our addressing system may not always be appropriate, however, is in cases where a transient amount of off-target expression would be detrimental. Because our system blocks transfer by degrading the message after it has entered the recipient cell, it is possible that genes on the message could be expressed in an off-target recipient before the message is

cleaved and degraded—indeed, some genes carried on the F plasmid have been observed to express as soon as 10 min after the plasmid's initial entrance into a receiver cell[55,56]. Preliminary experiments with our $F_{HR}$-based system, however, suggest that when genes are expressed weakly from the message plasmid, Cas9-mediated cleavage can occur quickly enough to prevent any detectable activity of these genes within off-target recipients (Supplementary Note; Supplementary Fig. 7). A thorough analysis of this phenomenon will likely require a comprehensive characterization of various transfer and blocking systems.

This transient expression phenomenon highlights the fact that DNA-based communication will not necessarily be the appropriate tool for every application. Indeed, although SMA channels do not exhibit many of the useful advantages of DNA channels, their simplicity and reliability nonetheless let them fill a valuable niche for the efficient implementation of low-complexity communication. In contrast, the role that DNA messaging is well suited to play in the continued development of consortium engineering is to push the boundaries of achievable complexity in the space of behaviors that can be programmed into a system.

One of the most promising examples of such a development is the integration of DNA messaging with the rapidly-advancing field of DNA writing and recording[57]. By allowing cells to directly pass the contents of their recordings to other cells without compression, DNA messaging can enable consortium-level actuation based on these recordings without the drawbacks of bottlenecks from low-capacity communication channels. DNA writing technologies could also be harnessed to generate biologically-interpretable messages de novo, paving the way for the types of fully autonomous self-reconfiguring systems that will enable new types of computation and actuation inaccessible to non-biological substrates[58].

By leveraging the dynamic mutability of DNA messages, alongside the message-channel decoupling and high information density already demonstrated by Ortiz and Endy, our work serves as a second step in the foundation of DNA messaging by creating a single generalizable system that embodies all three of its unique advantages. The ability to leverage a decade of intensive efforts to develop effective molecular DNA editors was critical in enabling our framework, and as these tools continue to advance, DNA messaging is itself poised to increase its functional capacity. Such future progress in DNA messaging that improves and expands upon the three advantages highlighted in our system will bring the field increasingly closer to realizing the ability to engineer autonomous, adaptive multicellular systems that rival the complexity of living systems.

## Methods

### Strain and plasmid construction

A list of all strains and constructs used in this study, associated with the experiments in which they were used, can be found in Supplementary Data 1. The parent strain of the Keio single-gene knockout collection[59], *E. coli* BW25113, was used as the basis for all experiments in this study with the exception of those described in Supplementary Figs. 6 (*E. coli* MG1655) and 7 (*E. coli* Marionette MG1655[60] for the receiver strains). Genomic integrations were performed using the pOSIP clonetegration system[61].

Because the $F_{HR}$ plasmid retains a low rate of self-transfer activity and carries a tetracycline resistance gene, the plasmid could be transferred from the original $F_{HR}$ donor strain into newly constructed sender strains using standard mating procedures (see below) and selectively plating for transconjugants.

All new plasmids for this study were constructed via 3G assembly[62] using genetic parts from the CIDAR MoClo extension part kit[63,64] when available. Parts not in the kit were converted to 3G-compatible parts by amplifying them from an existing source with custom primers or synthesizing the parts directly before combining them with the part plasmid backbone via Gibson assembly. The former approach was

used for the inducible promoters and their cognate regulators, taken from the Marionette system[60]; the spectinomycin and apramycin resistance genes, taken from the pQCascade and pCutAmp plasmids, respectively[65]; the gentamycin resistance cassette, taken from the pJM220 plasmid[66]; and the F *oriT* sequence, taken from the mobile GFP plasmid developed by Dimitriu et al.[18].

The latter synthesis-based approach was used for the gRNAs and address regions. Because the gRNAs target sites on the address region whose sequence can be fully specified by the user, one can, in principle, choose any arbitrary 20 bp sequence to serve as the recognition sequence of the gRNA. In order to avoid crosstalk with the *E. coli* genome, we chose 20 bp from the synthetically generated UNS2 sequence[67] to serve as the A site and 20 bp from the sequence of the yeast endonuclease I-SceI[68] to serve as the B site. The orthogonality of both sequences to the *E. coli* genome was validated with BLAST before construction. Address regions were constructed to include approximately 100 bp of spacer sequence between the actual gRNA binding site and the flanking integrase attachment site on each side. These spacer sequences were generated by taking random sequences from the interior of the ampicillin resistance gene, and their orthogonality to the gRNA sequences was validated with BLAST[69]. The D1–D4 gRNAs used in Fig. 4 correspond to those labeled "sequence 1" through "sequence 4", respectively, in Fig. 1 of Didovyk et al.[35].

All message plasmids in this study, with the exception of those in Supplementary Figure 7, were constructed on a low-copy pSC101-origin backbone. The message plasmids in Supplementary Fig. 7 were constructed on a high copy ColE1-origin backbone (Supplementary Note). All nonmobile plasmids used in this study were constructed on a medium-copy p15a-origin backbone.

### Cell culturing and plasmid transfer experiments

Strains involved in the transfer experiments were grown overnight in 2 mL of LB media in a 15 mL polypropylene culture tube in a shaking incubator (Thermo MaxQ 4000) set to 37 °C and 250 rpm under antibiotic selection for each resistance present in the strain. In the morning, each culture was diluted 1:100 into 2 mL of fresh LB media containing antibiotic selection for only the plasmid-based resistances in the strain and returned to the shaking incubator until the culture reached the mid-log phase (approx. 1–2 h). At this point, cultures were induced with the appropriate amount of inducer, if applicable, and continued incubating for another 1 h.

Cultures were then removed from the incubator, and their OD600 value was measured. One mL of the culture was then transferred into a 1.5 mL tube and spun at 1377g for 10 min on a tabletop centrifuge. The supernatant was removed, and the cell pellet was resuspended in 1 mL of fresh LB containing only kanamycin (or no antibiotics, for Supplementary Fig. 6), alongside the appropriate concentration of inducer if applicable. Cultures were then spun again at the same settings and resuspended in fresh media as before.

Cells in the washed cultures were then diluted into a single 3 mL culture of fresh LB, containing only the appropriate inducers and kanamycin (or no antibiotics, for Supplementary Fig. 6), in a 15 mL culture tube. Cells were added such that each strain would have an OD600 value of 0.002 within the final coculture, which typically involved at least a 1:100 dilution from the original monoculture. The coculture was then placed back into the shaking incubator, marking the beginning of the 6 h coculturing window.

The concentrations of the antibiotics in the media, when used, were 25 μg/mL kanamycin, 12.5 μg/mL chloramphenicol, 25 μg/mL apramycin, 25 μg/mL spectinomycin, 15 μg/mL gentamicin, 50 μg/mL carbenicillin, and 10 μg/mL tetracycline.

### Selective plating and measuring strain densities

After 6 h of coculturing (or at hourly time points, for Fig. 6c and Supplementary Fig. 6), the culture tube was removed from the incubator, and the density of each strain was assessed by selective plating of serial dilutions of the culture. 60 mm-wide LB agar plates containing the appropriate antibiotic markers (the unique genomically integrated resistance cassette for the strain as well as chloramphenicol to select for the message plasmid, when appropriate) were used for selection. For all experiments except those in Fig. 6 and Supplementary Fig. 6, serial dilutions were performed with 10-fold steps in 100 μL volumes, and four 5 μL spots of successive dilutions spanning the expected density were spread onto a single plate. For the remaining experiments, serial dilutions were performed with 100-fold steps in 1 mL volumes, and 100 μL of one or two dilutions were spread onto individual plates. Plates were incubated at 37 °C until the formation of colonies (12–24 h).

Strain densities, measured as colony-forming units per mL, were calculated by counting the number of colonies on the selection plates and multiplying by the appropriate dilution factor. When multiple dilution factors displayed growth, the dilution factor with the highest number of colonies that still remained countable (i.e., colonies were clearly discernible and separable) was used to calculate the density. Colonies were counted manually.

When fluorescent proteins were used to distinguish different colonies on the same selection plate, as in Fig. 4, plates were imaged on an Olympus MVX10 microscope using MicroManager version 2.0.0 with bandpass filters at 427/10–25 nm (TagBFP), 504/12–25 nm (sfYFP), and 589/15–25 nm (mScarlet-I).

### Calculating transfer rates

The transfer rate was calculated as $T/(S*R)$, where $T$ is the density of transconjugants, $S$ is the density of senders, and $R$ is the total density of receivers, which includes the receivers that have become transconjugants. Although this ratio is a standard measure of transfer rate used in the literature[70,71], we note that some recent works use this measure in a subtly different way, calculating the transfer rate as $T/(S*R)$ but not including the transconjugant density within $R$ term[72–74]. We chose to preserve the inclusion of the transconjugants in the $R$ term for two reasons. First, by defining $R$ as the total receiver density, experiments where transconjugants cannot be distinguished from receivers on receiver-selecting plates, such as those in Fig. 4, can be analyzed in the same way as experiments where this distinction can be made. Second, if the transfer process ever went to completion and all receivers were converted into transconjugants, the value of $T/(S*R)$ would not be infinite as it would be if $R$ were allowed to go to 0.

We preserved our definition of $R$ for the calculation of the fractional receiver conversion, $T/R$ (Supplementary Fig. 2). This value should range between 0 and 1.

### PCR assay for message plasmid identity

Colonies were picked and resuspended into 10 μL of M9 minimal media, of which 1 μL was placed into two separate 10 μL PCR reactions with primers designed to bind to the *oriT* and either the A or B gRNA binding site. Primer sequences were, for the unedited message plasmid (A site), CGCAGAATCCAAGCCG and CGGATAAAGTCACCAGAGGTG (with an annealing temperature of 64 °C) and for the edited message plasmid (B site), GGGATAACAGGGTAATC and GATAAAGTCACCA-GAGG (with an annealing temperature of 56 °C).

The number of PCR cycles was adjusted for each reaction against positive control colonies (cells containing a single message plasmid with either just the A site or just the B site on its address) to reduce the probability of observing false positives in the assay. The temperature program was 5 min at 98 °C followed by N cycles of 10 s at 98 °C, 30 s at the annealing temperature, and 20 s at 72 °C. After the cycles were completed, the reaction was kept at 72 °C for an additional 5 min before cooling down to 4 °C. N was 23 for the reaction targeting the unedited address, and N was 26 for the reaction targeting the edited address.

PCR samples were run on a gel and imaged on a UV imager, and the presence and absence of a band for each sample were determined by eye.

## Statistics and reproducibility

No statistical method was used to predetermine the sample size for the experiments. Data were only excluded from the analyses when a clear error in the experimental setup was noticed, in which case all data from that replicate were thrown out. Each biological replicate was sourced from a single randomly-selected colony from an overnight streak from a stock of the appropriate strain. This colony was then propagated in liquid media and used that day for every relevant condition within the experiment. Investigators were not blinded to allocation during experiments and outcome assessments.

The paired $t$ test was used for all statistical significance analyses in the text. We assumed the data were log-normally distributed and therefore log-transformed the data prior to performing the test in order to compare the difference in the geometric means of the compared samples. We note that the test also carries an assumption of equal variance between the conditions. Because of the small sample size ($n = 3$) in our comparisons, the validity of both of these assumptions is difficult to determine, and we encourage the reader to keep this point in mind as they assess our statements of statistical significance.

Jupyter notebooks that reproduce all of the analyses and plots presented in this paper are provided (see "Code availability").

## Reporting summary

Further information on research design is available in the Nature Portfolio Reporting Summary linked to this article.

## Data availability

A description of all of the strains and constructs created for this study, as well as the GenBank accession IDs associated with their annotated sequences, can be found in Supplementary Data 1. Source data are provided in this paper.

## Code availability

Jupyter notebooks (written for Python 3.8.8) that use the Source Data to reproduce all of the analyses and data figures in this manuscript can be found at github.com/jpmarken/DNAmessaging or https://doi.org/10.5281/zenodo.7700530[76].

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

## Acknowledgements

We thank A. Halleran, A. Shur, and R. Williams for insightful discussions, and the members of the Murray lab for feedback on the manuscript. We also thank T. Dimitriu for providing plasmids from the F$_{HR}$ system. Some of the schematic illustrations in this manuscript contain components adapted from the SBOL visual glyph library[75]. This material is based upon work supported by the National Science Foundation Graduate Research Fellowship Program under Grant no. DGE-1745301 and by the Institute

for Collaborative Biotechnologies through grants W911NF-09-D-0001 and W911NF-19-2-0026 from the U.S. Army Research Office. The content of the information in this report does not necessarily reflect the position or the policy of the Government, and no official endorsement should be inferred.

## Author contributions

J.P.M. conceived and designed the project, performed the experiments, analyzed the data, and wrote the paper, all under the supervision of R.M.M.

## Competing interests

The authors declare no competing interests.
