## [Peer Review File · Nature Communications]

Reviewers' Comments:

Reviewer #1:

Remarks to the Author:

The authors demonstrated that conjugation and Cas9-mediated cleavage can be used to target specific receiver strains and that serine integrases can be used to edit this message in situ to control the propagation of the signal through the coculture. We find that the system described here has potential for scaling up communication channels for microbial consortia, which currently rely on the transfer of small molecules such as quorum sensing molecules or antibiotics to control the flow of information in the coculture. The manuscript is well-written and can be accepted without additional experiments.

Other Comments:

Introduction – line 39

Authors should add the following references of molecules used for communication in microbial consortia:

1) Fedorec, Alex JH, et al. "Single strain control of microbial consortia." *Nature communications* 12.1 (2021): 1-12.

2) Kong, Wentao, et al. "Designing microbial consortia with defined social interactions." *Nature Chemical Biology* 14.8 (2018): 821-829.

Figure 4

Could the authors precise how the gRNAs were designed and if multiple address regions were tested to reduce the off-target effects between the different receiver strains.

Line 225

"We then performed performed [...]" - typo

Methods - line 350

Authors should explain why they chose a high copy origin of replication to build the message plasmid in Figure S7, while they use a low copy ori in the rest of the paper. If the copy number of the message plasmids affects the transfer rate of the system, then they should provide experimental evidence to support their answer.

Reviewer #2:

Remarks to the Author:

Murken and Murray present a manuscript entitled "Addressable and adaptable intercellular communication via DNA messaging" where the use of plasmid conjugation is applied in microbial communities.

The experiments are solid as well as the results. The experimental design is well thought and through simple experiments, the authors demonstrate their points. The manuscript is very clear and well-written.

Comments:

-The intro should be expanded to discuss relevant research that has been done using DNA conjugation in engineering microbiomes/communities.

-The fact that this work is done in *E. coli* should be mentioned in the abstract and introduction. This is only mentioned in the methods.

-One potential reason why there has not been too much research on this topic specifically, could be that the type of applications that are enabled by this technology is challenged by its limitations. I somehow find it difficult to see how these technologies can be transformed into applications, especially when only a fraction of the population is conjugated and there is certain leakiness. It would be nice if the authors describe or even demonstrate some examples of applications.

-Why do the authors think that D3 gRNA is less effective than the rest? Can these rates be improved by screening a larger number of gRNAs?

-In the experiments with the 3 fluorescent proteins, it would be great to expand the methods with

the specific filters that were used, which is important since there are some overlaps of spectra.

-For the experiment depicted in figure 6c, it seems that with time, the populations of T2 and T3 are very similar, why do the authors think that the rate of conjugation for T3 is higher than T2? Is this expected and how could it be avoided?

-Line 259, are the differences mentioned in the text (5 fold) significant? It would be great to add statistics more generally in all graphs to determine if the observed differences are statistically significant overall.

-Because the transfer rates are not calculated in the standard manner, it may be good that this is indicated earlier in the text or figure captions and not only in the method section. I would indicate throughout the text the % of cells that are actually conjugated to have an estimation of the heterogeneity of the final populations.

-The section "Address editing enables control of information flow through a population" is thought to be a proof of concept of the tools described in this manuscript. I however miss some more information on why the experiment was designed like this and not in other potential configurations. What type of applications could benefit from the generated behaviour?

Summary of Revisions

Main Text

- Typos have been fixed.
- Some additional references have been added.
- The phrasing of some statements have been modified for clarity.
- Where italics were previously used to indicate emphasis, they have been unitalicized.
- Figure titles have been bolded.
- The discussion of Fig 5's results has been rewritten to increase clarity and incorporate reviewer suggestions.
- Most of the section describing the Fig 6 experiments has been rewritten to increase clarity and incorporate reviewer suggestions.
- We have corrected an error in Fig 6b where the height of the bar graphs was showing the arithmetic mean, rather than the geometric mean, of the strain densities.
- A paragraph was added to the Discussion to comment on potential applications of DNA messaging.
- A more detailed description of the construction of the gRNA and address region sequences has been added to the Methods.
- A more detailed explanation of our calculation of the Transfer Rate has been added to the Methods.
- The strain background for Fig S5 (now S6) was incorrectly indicated as JS006 in the Methods. It has been corrected to MG1655.
- The Data Availability section has been modified to include the availability

Supplemental Materials

- The figure formerly labeled S6 has been moved to the position of S2, shifting the other figures down correspondingly. The relabeling is described by the following table.

Old Label	New Label
S1	S1
S2	S3
S3	S4
S4	S5
S5	S6
S6	S2
S7	S7

- Additional graphs have been added to Figs S2,S3,S6 (now S3,S4,S2) to show additional views of the data, particularly the percent of receivers converted to transconjugants.
- Error bars have been added to the correlation plot in Fig S3 (now S4)
- A new section, Supplemental Note, has been added to provide greater contextualization of the Fig S7 experiment.
- Figure captions have been rewritten in places for increased clarity.
- The associated jupyter notebooks to generate the graphs have been updated accordingly.
- The figure labels in the Supplemental Data Files have been updated to match the new labels in the text.

Response to *Reviewers' Comments*

Reviewer #1:

The authors demonstrated that conjugation and Cas9-mediated cleavage can be used to target specific receiver strains and that serine integrases can be used to edit this message in situ to control the propagation of the signal through the coculture. We find that the system described here has potential for scaling up communication channels for microbial consortia, which currently rely on the transfer of small molecules such as quorum sensing molecules or antibiotics to control the flow of information in the coculture. The manuscript is well-written and can be accepted without additional experiments.

- We thank the reviewer for their assessment.

Other Comments:

Introduction – line 39

Authors should add the following references of molecules used for communication in microbial consortia:

- 1) Fedorec, Alex JH, et al. "Single strain control of microbial consortia." *Nature communications* 12.1 (2021): 1-12.
- 2) Kong, Wentao, et al. "Designing microbial consortia with defined social interactions." *Nature Chemical Biology* 14.8 (2018): 821-829.

- We have added the designated references.

Figure 4

Could the authors precise how the gRNAs were designed and if multiple address regions were tested to reduce the off-target effects between the different receiver strains.

- We have expanded our discussion of the gRNA construction in the Methods. The two gRNA sequences used throughout the paper ("A" and "B") were the first sequences designed, and since they showed good orthogonality to each other in initial experiments, we did not design any additional sequences to further optimize the crosstalk.

Line 225

"We then performed performed [...]" – typo

- We have corrected the error.

Methods - line 350

Authors should explain why they chose a high copy origin of replication to build the message plasmid in Figure S7, while they use a low copy ori in the rest of the paper. If the copy number of the message plasmids affects the transfer rate of the system, then they should provide experimental evidence to support their answer.

- We thank the reviewer for pointing out this lack of clarity in our initial description of the experiment. We used a high-copy origin of replication for this experiment in order to ensure that both the weakly-expressing and strongly-expressing integrase constructs would express a sufficient quantity of integrase to activate the antibiotic resistance cassette on the genome, as a preliminary test of the weakly-expressing integrase construct on a low-copy backbone failed to activate the resistance phenotype. We have included this justification in the new Supplemental Note section that describes the motivation and design of the Fig S7 experiment in more detail.

As to the impact of the copy number on the transfer rate, we did not perform a rigorous investigation of this property because doing so would require a thorough comparison of many different plasmid backbones that we felt was beyond the scope of this study. We would like to emphasize, however, that the inclusion of the YFP plasmid condition in Figure S7 acts as a standard that allows us to normalize the transfer rates of the two integrase plasmids in such a way that cancels out any global changes in transfer rate between the high-copy origin used in this experiment and the low-copy origin used in the rest of the paper. The use of the high-copy origin in Figure S7 does not, therefore, compromise the conclusions drawn from its results.

Reviewer #2:

Murken and Murray present a manuscript entitled “Addressable and adaptable intercellular communication via DNA messaging” where the use of plasmid conjugation is applied in microbial communities.

The experiments are solid as well as the results. The experimental design is well thought and through simple experiments, the authors demonstrate their points. The manuscript is very clear and well-written.

- We thank the reviewer for their assessment.

Comments:

-The intro should be expanded to discuss relevant research that has been done using DNA conjugation in engineering microbiomes/communities.

- We have modified the text of the introduction to place more emphasis on other work by synthetic biologists to engineer environmental microbiomes using horizontal gene transfer (including conjugation). We did not choose to elaborate too extensively on these studies, however, as we feel that the focus of the introduction should be on discussing DNA transfer as a communication framework for engineered consortia, rather than as a means of performing *in situ* engineering of intact native microbiomes.

-The fact that this work is done in E. coli should be mentioned in the abstract and introduction. This is only mentioned in the methods.

- We thank the reviewer for catching this oversight. We have modified the abstract to state that this work is done in *E. coli*, and we have also restated this in the main text when the first experimental results (Fig 3) are introduced.

-One potential reason why there has not been too much research on this topic specifically, could be that the type of applications that are enabled by this technology is challenged by its limitations. I somehow find it difficult to see how these technologies can be transformed into applications, especially when only a fraction of the population is conjugated and there is certain leakiness. It would be nice if the authors describe or even demonstrate some examples of applications.

- This is an excellent suggestion. One of the most promising areas for immediate application of this system is to interface cell-cell communication with the emerging field of DNA recording. Currently, DNA recording systems cannot effectively interface with engineered consortia because the recording cell faces an information bottleneck when trying to communicate its memory to another cell. Our system alleviates this bottleneck by allowing the recording cell to directly transfer the DNA memory to another cell in the consortium, enabling the division of labor between sensing and actuation in the system. More generally, the ability for DNA messages to dynamically and autonomously edit themselves makes them well-suited for implementing adaptable, self-reconfiguring multicellular systems. A paragraph on these thoughts has been added to the Discussion section.

-Why do the authors think that D3 gRNA is less effective than the rest? Can these rates be improved by screening a larger number of gRNAs?

- We do not have any ideas as to why the D3 gRNA is less effective than the others, particularly since its ineffectiveness is not universal— the D3 gRNA is able to block a message plasmid that contains all three binding sites (D1, D2, and D3) with comparable effectiveness to the D1 and D2 gRNAs, as can be seen in Fig 4b and Fig S2. If we had to guess, there is likely some interaction between the D3 binding site and another site on the message plasmid that is ameliorated in the D1D2D3 plasmid— but we have chosen not to include such speculations in the main text as we have no evidence to support these claims.
Although screening additional gRNAs would almost certainly yield better candidates than D3, we note that the D1 and D2 gRNAs already led to an undetectable level (<200 CFUs/mL) of transconjugants in some conditions (Fig S2). Further improving on these gRNAs' performance would therefore require a screen that uses a more sensitive assay for transfer blocking.

-In the experiments with the 3 fluorescent proteins, it would be great to expand the methods with the specific filters that were used, which is important since there are some overlaps of spectra.

- We thank the reviewer for catching this oversight— this information is now included in the Methods.

-For the experiment depicted in figure 6c, it seems that with time, the populations of T2 and T3 are very similar, why do the authors think that the rate of conjugation for T3 is higher than T2? Is this expected and how could it be avoided?

- In the timecourse data presented in Fig 6c, the density of the Strain T3 population does indeed reach a level that is very close to the density of the Strain T2 population. However, it is also the case that the density of T3's parent strain, Strain 3, is notably higher than that of T2's parent strain, Strain 2 (see Fig 6b). The observed convergence of the T3 trajectory to the T2 trajectory is therefore a function of differences in both the growth rate of the base strains and in the transfer rate into the two strains. When the contribution of the transfer rate is isolated in Fig 6e by normalizing out the growth differences between the strains, we see that the transfer rates to T2 and T3 are quite similar, as expected (geometric mean of $4e-10$ vs. $5e-10$). The observed convergence in Fig 6c is therefore due primarily to the faster growth rate of T3 compared to T2. In order to avoid this phenomenon, care should be taken to ensure that each constituent strain in the population has a similar effective growth rate. This could be accomplished either by active feedback mechanisms or by designing the strains to experience a similar level of gene expression burden.

-Line 259, are the differences mentioned in the text (5 fold) significant? It would be great to add statistics more generally in all graphs to determine if the observed differences are statistically significant overall.

- Throughout the paper we have chosen to not include statistical analyses of significance, instead choosing to show the direct values of each data point in the graphs so that readers can make their own assessments about whether they agree with the claims we draw from our data. We felt that the inclusion of statements about statistical significance could give the reader a false sense of confidence in our conclusions that is not warranted by our small sample sizes, and so we chose to simply present the data as-is to encourage a skeptical assessment of our results.

With respect to the specific claim in line 259, our hope is that readers would see our two stated claims (that the density of Strain T3 was not higher than Strain T2, and that the density of Strain 3 was higher than Strain 2) and then look at the data in Fig 6b to determine whether they feel the data supports these conclusions. The inclusion of a statement of statistical significance within the text that implicitly justifies these claims, however, might lead some readers to accept them at face value without examining the actual data in Fig 6b.

We therefore hope that the reviewer will accept our wish to present our data purely at face value.

-Because the transfer rates are not calculated in the standard manner, it may be good that this is indicated earlier in the text or figure captions and not only in the method section. I would indicate throughout the text the % of cells that are actually conjugated to have an estimation of the heterogeneity of the final populations.

- We have clarified the calculation of the Transfer Rate in the caption of Fig 3 where it first appears, and have also expanded our discussion of the Transfer Rate measure in the Methods section to more clearly explain the context behind, and justification for, our choice of transfer rate measure.

We have included within the text the percentage of receivers that have been converted to transconjugants within the 6-hour coculturing window for all experimental figures, and graphs showing these data have also been added to the appropriate supplemental figures.

-The section “Address editing enables control of information flow through a population” is thought to be a proof of concept of the tools described in this manuscript. I however miss some more information on why the experiment was designed like this and not in other potential configurations. What type of applications could benefit from the generated behaviour?

- We chose to demonstrate our system’s capabilities in a linear relay system because we felt that it acted as a minimal demonstration of the enforcing of a sequential ordering within a process. Enforced ordering is a fundamental part of the operation of any multi-step process, whether it be a metabolic pathway, manufacturing pipeline, or even a computational algorithm. If a consortium is engineered in order to assemble some product, then ensuring that the multiple steps in that assembly process occur in their defined order will likely be an important part of its function, and ensuring that the instructions governing the performance of these steps can themselves be passed in a sequential ordering would be a good way to implement this. We have included an expanded discussion along these lines to the main text at the beginning of this section.

Reviewers' Comments:

Reviewer #1:

Remarks to the Author:

All my concerns have been addressed and the authors have made the necessary changes to clarify the text and figures where needed.

Reviewer #2:

Remarks to the Author:

The authors have partially addressed my comments.

My main comment now is the decision to avoid statistical analysis of the data under the premise that they don't want the reader to make their own conclusions on the significance. I think that is not a common or good practise and I think it would be better to include the statistical analysis and then discuss the limitations that are mentioned in the response document, so the reader is aware of those. This is just my opinion, but the editor will probably know better what is expected in this journal.

Reviewer #1 (Remarks to the Author):

All my concerns have been addressed and the authors have made the necessary changes to clarify the text and figures where needed.

We thank the reviewer for their productive feedback.

Reviewer #2 (Remarks to the Author):

The authors have partially addressed my comments.

My main comment now is the decision to avoid statistical analysis of the data under the premise that they don't want the reader to make their own conclusions on the significance. I think that is not a common or good practise and I think it would be better to include the statistical analysis and then discuss the limitations that are mentioned in the response document, so the reader is aware of those. This is just my opinion, but the editor will probably know better what is expected in this journal.

We thank the reviewer for their productive feedback. We have now performed statistical analyses to assess the significance of comparisons made in the text associated with Figures 3, 5, and 6. We have included the results of these statistical tests in the main text where the comparisons are made and have included a general discussion on the statistics and their limitations in the Statistics and Reproducibility subsection of the Methods.